# BWCP: Probabilistic Learning-to-Prune Channels for ConvNets via Batch Whitening

## Abstract

This work presents a probabilistic channel pruning method to accelerate Convolutional Neural Networks (CNNs). Previous pruning methods often zero out unimportant channels in training in a deterministic manner, which reduces CNN's learning capacity and results in suboptimal performance. To address this problem, we develop a probability-based pruning algorithm, called batch whitening channel pruning (BWCP), which can stochastically discard unimportant channels by modeling the probability of a channel being activated. BWCP has several merits. (1) It simultaneously trains and prunes CNNs from scratch in a probabilistic way, exploring larger network space than deterministic methods. (2) BWCP is empowered by the proposed batch whitening tool, which is able to empirically and theoretically increase the activation probability of useful channels while keeping unimportant channels unchanged without adding any extra parameters and computational cost in inference. (3) Extensive experiments on CIFAR-10, CIFAR-100, and ImageNet with various network architectures show that BWCP outperforms its counterparts by achieving better accuracy given limited computational budgets. For example, ResNet50 pruned by BWCP has only 0.58% Top-1 accuracy drop on ImageNet, while reducing 42.9% FLOPs of the plain ResNet50.

## 1 Introduction

Deep convolutional neural networks (CNNs) have achieved superior performance in a variety of computer vision tasks such as image recognition (He et al., 2016), object detection (Ren et al., 2017), and semantic segmentation (Chen et al., 2018). However, despite their great success, deep CNN models often have massive demand on storage, memory bandwidth, and computational power (Han & Dally, 2018), making them difficult to be plugged onto resource-limited platforms, such as portable and mobile devices (Deng et al., 2020). Therefore, proposing efficient and effective model compression methods has become a hot research topic in the deep learning community.

Model pruning, as one of the vital model compression techniques, has been extensively investigated. It reduces model size and computational cost by removing unnecessary or unimportant weights or channels in a CNN (Han et al., 2016). For example, many recent works (Wen et al., 2016; Guo et al., 2016) prune fine-grained weights of filters. Han et al. (2015) proposes to discard the weights that have magnitude less than a predefined threshold. Guo et al. (2016) further utilizes a sparse mask on a weight basis to achieve pruning. Although these unstructured pruning methods achieve optimal pruning schedule, they do not take the structure of CNNs into account, preventing them from being accelerated on hardware such as GPU for parallel computations (Liu et al., 2018).

To achieve efficient model storage and computations, we focus on structured channel pruning (Wen et al., 2016; Yang et al., 2019a; Liu et al., 2017), which removes entire structures in a CNN such as filter or channel. A typical structured channel pruning approach commonly contains three stages, including pre-training a full model, pruning unimportant channels by the predefined criteria such as $\ell_p$ norm, and fine-tuning the pruned model (Liu et al., 2017; Luo et al., 2017), as shown in Fig.1 (a). However, it is usually hard to find a global pruning threshold to select unimportant channels, because the norm deviation between channels is often too small (He et al., 2019). More importantly, as some channels are permanently zeroed out in the pruning stage, such a multi-stage procedure usually not only relies on hand-crafted heuristics but also limits the learning capacity (He et al., 2018a; 2019).

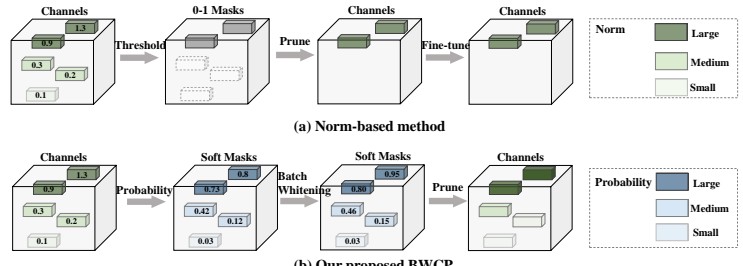

Figure 1: Illustration of our proposed BWCP. (**a**) Previous channel pruning methods utilize a hard criterion such as the norm (Liu et al., 2017) of channels to deterministically remove unimportant channels, which deteriorates performance and needs a extra fine-tuning process(Frankle & Carbin, 2018). (**b**) Our proposed BWCP is a probability-based pruning framework where unimportant channels are stochastically pruned with activation probability, thus maintaining the learning capacity of original CNNs. In particular, our proposed batch whitening (BW) tool can increase the activation probability of useful channels while keeping the activation probability of unimportant channels unchanged, enabling BWCP to identify unimportant channels reliably.

To tackle the above issues, we propose a simple but effective probability-based channel pruning framework, named batch-whitening channel pruning (BWCP), where unimportant channels are pruned in a stochastic manner, thus preserving the channel space of CNNs in training (*i.e.* the diversity of CNN architectures is preserved). To be specific, as shown in Fig.1 (b), we assign each channel with an activation probability (*i.e.* the probability of a channel being activated), by exploring the properties of the batch normalization layer (Ioffe & Szegedy, 2015; Arpit et al., 2016). A larger activation probability indicates that the corresponding channel is more likely to be preserved.

We also introduce a capable tool, termed batch whitening (BW), which can increase the activation probability of useful channels, while keeping the unnecessary channels unchanged. By doing so, the deviation of the activation probability between channels is explicitly enlarged, enabling BWCP to identify unimportant channels during training easily. Such an appealing property is justified by theoretical analysis and experiments. Furthermore, we exploit activation probability adjusted by BW to generate a set of differentiable masks by a soft sampling procedure with Gumbel-Softmax technique, allowing us to train BWCP in an online "pruning-from-scratch" fashion stably. After training, we obtain the final compact model by directly discarding the channels with zero masks.

The main **contributions** of this work are three-fold. (1) We propose a probability-based channel pruning framework BWCP, which explores a larger network space than deterministic methods. (2) BWCP can easily identify unimportant channels by adjusting their activation probabilities without adding any extra model parameters and computational cost in inference. (3) Extensive experiments on CIFAR-10, CIFAR-100 and ImageNet datasets with various network architectures show that BWCP can achieve better recognition performance given the comparable amount of resources compared to existing approaches. For example, BWCP can reduce 68.08% Flops by compressing 93.12% parameters of VGG-16 with merely accuracy drop and ResNet-50 pruned by BWCP has only 0.58% top-1 accuracy drop on ImageNet while reducing 42.9% FLOPs.

## 2 RELATED WORK

**Weight Pruning.** Early network pruning methods mainly remove the unimportant weights in the network. For instance, Optimal Brain Damage (LeCun et al., 1990) measures the importance of weights by evaluating the impact of weight on the loss function and prunes less important ones. However, it is not applicable in modern network structure due to the heavy computation of the Hessian matrix. Recent work assesses the importance of the weights through the magnitude of the weights itself. Specifically, (Guo et al., 2016) prune the network by encouraging weights to become exactly zero. The computation involves weights with zero can be discarded. However, a major drawback of weight pruning techniques is that they do not take the structure of CNNs into account, thus failing to help scale pruned models on commodity hardware such as GPUs (Liu et al., 2018; Wen et al., 2016).

**Channel Pruning.** Channel pruning approaches directly prune feature maps or filters of CNNs, making it easy for hardware-friendly implementation. For instance, relaxed $\ell_0$ regularization (Louizos

et al., 2017) and group regularizer (Yang et al., 2019a) impose channel-level sparsity, and filters with small value are selected to be pruned. Some recent work also propose to rank the importance of filters by different criteria including $\ell_1$ norm (Liu et al., 2017; Li et al., 2017), $\ell_2$ norm (Frankle & Carbin, 2018) and High Rank channels (Lin et al., 2020). For example, (Liu et al., 2017) explores the importance of filters through scale parameter $\gamma$ in batch normalization. Although these approaches introduce minimum overhead to the training process, they are not trained in an end-to-end manner and usually either apply on a pre-trained model or require an extra fine-tuning procedure.

Recent works tackle this issue by pruning CNNs from scratch. For example, FPGM (He et al., 2019) zeros in unimportant channels and continues training them after each training epoch. Furthermore, both SSS and DSA learn a differentiable binary mask that is generated by channel importance and does not require any additional fine-tuning. Our proposed BWCP is most related to variational pruning (Zhao et al., 2019) and SCP (Kang & Han, 2020) as they also employ the property of normalization layer and associate the importance of channel with probability. The main difference is that our method adopts the idea of whitening to perform channel pruning. We will show that the proposed batch whitening (BW) technique can adjusts the activation probability of different channels according to their importance, making it easy to identify unimportant channels. Although previous work SPP(Wang et al., 2017) and DynamicCP (Gao et al., 2018) also attempt to boost salient channels and skip unimportant ones, they fail to consider the natural property inside normalization layer and deign the activation probability empirically .

## 3 PRELIMINARY

**Notation.** We use regular letters, bold letters, and capital letters to denote scalars such as '$x$', and vectors (*e.g.* vector, matrix, and tensor) such as '$\mathbf{x}$' and random variables such as '$X$', respectively.

We begin with introducing a building layer in recent deep neural nets which typically consists of a convolution layer, a batch normalization (BN) layer, and a rectified linear unit (ReLU) (Ioffe & Szegedy, 2015; He et al., 2016). Formally, it can be written by

$$\mathbf{x}_c = \mathbf{w}_c * \mathbf{z}, \quad \tilde{\mathbf{x}}_c = \gamma_c \bar{\mathbf{x}}_c + \beta_c, \quad \mathbf{y}_c = \max\{\mathbf{0}, \tilde{\mathbf{x}}_c\} \tag{1}$$

where $c \in [C]$ denotes channel index and $C$ is channel size. In Eqn.(1), '$*$' indicates convolution operation and $\mathbf{w}_c$ is filter weight corresponding to the $c$-th output channel, *i.e.* $\mathbf{x}_c \in \mathbb{R}^{N \times H \times W}$. To perform normalization, $\mathbf{x}_c$ is firstly standardized to $\bar{\mathbf{x}}_c$ through $\bar{\mathbf{x}}_c = (\mathbf{x}_c - \mathbb{E}[\mathbf{x}_c])/\sqrt{\mathbb{D}[\mathbf{x}_c]}$ where $\mathbb{E}[\cdot]$ and $\mathbb{D}[\cdot]$ indicate calculating mean and variance over a batch of samples, and then is re-scaled to $\tilde{\mathbf{x}}_c$ by scale parameter $\gamma_c$ and bias $\beta_c$. Moreover, the output feature $\mathbf{y}_c$ is obtained by ReLU activation that discards the negative part of $\tilde{\mathbf{x}}_c$.

**Criterion-based channel pruning.** For channel pruning, previous methods usually employ a 'small-norm-less-important' criterion to measure the importance of channels. For example, BN layer can be applied in channel pruning (Liu et al., 2017), where a channel with a small value of $\gamma_c$ would be removed. The reason is that the $c$-th output channel $\tilde{\mathbf{x}}_c$ contributes little to the learned feature representation when $\gamma_c$ is small. Hence, the convolution in Eqn.(1) can be discarded safely, and filter $\mathbf{w}_c$ can thus be pruned. Unlike these criterion-based methods that deterministically prune unimportant filters and rely on a heuristic pruning procedure as shown in Fig.1(a), we explore a probability-based channel pruning framework where less important channels are pruned in a stochastic manner.

**Activation probability.** To this end, we define an activation probability of a channel by exploring the property of the BN layer. Those channels with a larger activation probability could be preserved with a higher probability. To be specific, since $\bar{\mathbf{x}}_c$ is acquired by subtracting the sample mean and being divided by the sample variance, we can treat each channel feature as a random variable following standard Normal distribution (Arpit et al., 2016), denoted as $\bar{X}_c$. Note that only positive parts can be activated by ReLU function. Proposition 1 gives the activation probability of the $c$-th channel, *i.e.* $P(\tilde{X}_c) > 0$.

**Proposition 1** *Let a random variable $\bar{X}_c \sim \mathcal{N}(0,1)$ and $Y_c = max\{0, \gamma_c \bar{X}_c + \beta_c\}$. Then we have (1) $P(Y_c > 0) = P(\tilde{X}_c > 0) = (1 + \mathrm{Erf}(\beta_c/(\sqrt{2}|\gamma_c|)))/2$ where $\mathrm{Erf}(x) = \int_0^x 2/\sqrt{\pi} \cdot \exp^{-t^2} dt$, and (2) $P(\tilde{X}_c > 0) = 0 \Leftrightarrow \beta_c \leq 0$ and $\gamma_c \to 0$.*

Note that a pruned channel can be modelled by $P(\tilde{X}_c > 0) = 0$. With Proposition 1 (see proof in Appendix A.2), we know that the unnecessary channels satisfy that $\gamma_c$ approaches 0 and $\beta_c$ is

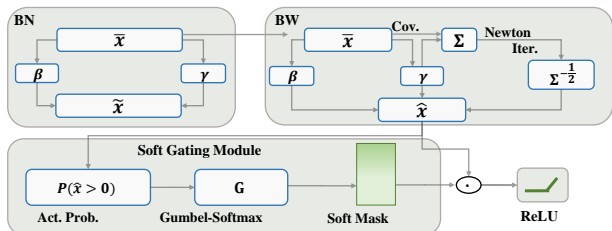

Figure 2: A schematic of the proposed Batch Whitening Channel Pruning (BWCP) algorithm that consists of a BW module and a soft sampling procedure. By modifying BN layer with a whitening operator, the proposed BW technique adjusts activation probabilities of different channels. These activation probabilities are then utilized by a soft sampling procedure.

negative. To achieve channel pruning, previous compression techniques (Li et al., 2017; Zhao et al., 2019) merely impose a regularization on $\gamma_c$, which would deteriorate the representation power of unpruned channels (Perez et al., 2018; Wang et al., 2020). Instead, we adopt the idea of whitening to build a probabilistic channel pruning framework where unnecessary channels are stochastically disgarded with a small activation probability while important channels are preserved with a large activation probability.

# 4 BATCH WHITENING CHANNEL PRUNING

This section introduces the proposed batch whitening channel pruning (BWCP) algorithm, which contains a batch whitening module that can adjust the activation probability of channels, and a soft sampling module that stochastically prunes channels with the activation probability adjusted by BW. The whole pipeline of BWCP is illustrated in Fig.2.

By modifying the BN layer in Eqn.(1), we have the formulation of BWCP,

$$\mathbf{x}_c^{\text{out}} = \underbrace{\hat{\mathbf{x}}_c}_{\text{batch whitening}} \odot \underbrace{m_c(P(\hat{X}_c > 0))}_{\text{soft sampling}} \tag{2}$$

where $\mathbf{x}_c^{out}, \hat{\mathbf{x}}_c \in \mathbb{R}^{N \times H \times W}$ denote the output of proposed BWCP algorithm and BW module, respectively. '$\odot$' denotes broadcast multiplication. $m_c \in [0, 1]$ denotes a soft sampling that takes the activation probability of output features of BW (*i.e.* $P(\hat{X}_c > 0)$) and returns a soft mask. The closer the activation probability is to 0 or 1, the more likely the mask is to be hard. To distinguish important channels from unimportant ones, BW is proposed to increase the activation probability of useful channels while keeping the probability of unimportant channels unchanged during training. Since Eqn.(2) always retain all channels in the network, our BWCP can preserve the learning capacity of the original network during training (He et al., 2018a). The following sections present BW and soft sampling module in detail.

## 4.1 BATCH WHITENING

Unlike previous works (Zhao et al., 2019; Kang & Han, 2020) that simply measure the importance of channels by parameters in BN layer, we attempt to whiten features after BN layer by the proposed BW module. We show that BW can change the activation probability of channels according to their importances without adding additional parameters and computational overhead in inference.

As shown in Fig.2, BW acts after the BN layer. By rewriting Eqn.(1) into a vector form, we have the formulation of BW,

$$\hat{\mathbf{x}}_{nij} = \mathbf{\Sigma}^{-\frac{1}{2}}(\boldsymbol{\gamma} \odot \bar{\mathbf{x}}_{nij} + \boldsymbol{\beta}) \tag{3}$$

where $\hat{\mathbf{x}}_{nij} \in \mathbb{R}^{C \times 1}$ is a vector of $C$ elements that denote the output of BW for the $n$-th sample at location $(i, j)$ for all channels. $\mathbf{\Sigma}^{-\frac{1}{2}}$ is a whitening operator and $\mathbf{\Sigma} \in \mathbb{R}^{C \times C}$ is the covariance matrix of channel features $\{\tilde{\mathbf{x}}_c\}_{c=1}^C$. Moreover, $\boldsymbol{\gamma} \in \mathbb{R}^{C \times 1}$ and $\boldsymbol{\beta} \in \mathbb{R}^{C \times 1}$ are two vectors by stacking $\gamma_c$ and $\beta_c$ of all the channels respectively. $\bar{\mathbf{x}}_{nij} \in \mathbb{R}^{C \times 1}$ is a vector by stacking elements from all channels of $\bar{x}_{ncij}$ into a column vector.

**Training and inference.** Note that BW in Eqn.(3) requires computing a root inverse of a covariance matrix of channel features after the BN layer. Towards this end, we calculate the covariance matrix $\mathbf{\Sigma}$ within a batch of samples during each training step as given by

$$\boldsymbol{\Sigma} = \frac{1}{NHW} \sum_{n,i,j=1}^{N,H,W} (\boldsymbol{\gamma} \odot \bar{\mathbf{x}}_{nij})(\boldsymbol{\gamma} \odot \bar{\mathbf{x}}_{nij})^\mathsf{T} = (\boldsymbol{\gamma}\boldsymbol{\gamma}^\mathsf{T}) \odot \boldsymbol{\rho} \quad (4)$$

where $\boldsymbol{\rho}$ is a C-by-C correlation matrix of channel features $\{\bar{\mathbf{x}}_c\}_{c=1}^C$ (see details in Appendix A.1). The Newton Iteration is further employed to calculate its root inverse, $\boldsymbol{\Sigma}^{-\frac{1}{2}}$, as given by the following iterations

$$\boldsymbol{\Sigma}_k = \frac{1}{2}(3\boldsymbol{\Sigma}_{k-1} - \boldsymbol{\Sigma}_{k-1}^3 \boldsymbol{\Sigma}), \ k = 1, 2, \cdots, T. \quad (5)$$

where $k$ and $T$ are the iteration index and iteration number respectively and $\boldsymbol{\Sigma}_0 = \mathbf{I}$ is a identity matrix. Note that when $\|\mathbf{I} - \boldsymbol{\Sigma}\|_2 < 1$, Eqn.(5) converges to $\boldsymbol{\Sigma}^{-\frac{1}{2}}$ (Bini et al., 2005). To satisfy this condition, $\boldsymbol{\Sigma}$ can be normalized by $\boldsymbol{\Sigma}/\mathrm{tr}(\boldsymbol{\Sigma})$ following (Huang et al., 2019), where $\mathrm{tr}(\cdot)$ is the trace operator. In this way, the normalized covariance matrix can be written as $\boldsymbol{\Sigma}_N = \boldsymbol{\gamma}\boldsymbol{\gamma}^\mathsf{T} \odot \boldsymbol{\rho}/\|\boldsymbol{\gamma}\|_2^2$.

During inference, we use the moving average to calculate the population estimate of $\hat{\boldsymbol{\Sigma}}_N^{-\frac{1}{2}}$ by following the updating rules, $\hat{\boldsymbol{\Sigma}}_N^{-\frac{1}{2}} = (1-g)\hat{\boldsymbol{\Sigma}}_N^{-\frac{1}{2}} + g\boldsymbol{\Sigma}_N^{-\frac{1}{2}}$. Here $\boldsymbol{\Sigma}_N$ is the covariance calculated within each mini-batch at each training step, and $g$ denotes the momentum of moving average. Note that $\hat{\boldsymbol{\Sigma}}_N^{-\frac{1}{2}}$ is fixed during inference, the proposed BW does not introduce extra costs in memory or computation since $\hat{\boldsymbol{\Sigma}}_N^{-\frac{1}{2}}$ can be viewed as a convolution kernel with size of 1, which can be absorbed into previous convolutional layer. For completeness, we also analyze the training overhead of BWCP in Appendix Sec.A.3 where we see BWCP introduces a little extra training overhead.

## 4.2 Analysis of BWCP

In this section, we show that BWCP can easily identify unimportant channels by increasing the difference of activation between important and unimportant channels.

**Proposition 2** *Let a random variable $\bar{X} \sim \mathcal{N}(0,1)$ and $Y_c = max\{0, [\hat{\boldsymbol{\Sigma}}_N^{-\frac{1}{2}}(\boldsymbol{\gamma} \odot \bar{X} + \boldsymbol{\beta})]_c\}$. Then we have $P(Y_c > \delta) = P(\hat{X}_c > \delta) = (1 + \mathrm{Erf}((\hat{\beta}_c - \delta)/(\sqrt{2}|\hat{\gamma}_c|))/2$, where $\delta$ is a small positive constant, $\hat{\gamma}_c$ and $\hat{\beta}_c$ are two equivalent scale parameter and bias defined by BW module. Take $T = 1$ in Eqn.(5) as an example, we have $\hat{\gamma}_c = \frac{1}{2}(3\gamma_c - \sum_{d=1}^C \gamma_d^2\gamma_c\rho_{dc}/\|\boldsymbol{\gamma}\|_2^2)$, and $\hat{\beta}_c = \frac{1}{2}(3\beta_c - \sum_{d=1}^C \beta_d\gamma_d\gamma_c\rho_{dc}/\|\boldsymbol{\gamma}\|_2^2)$ where $\rho_{dc}$ is the Pearson's correlation between channel features $\bar{\mathbf{x}}_c$ and $\bar{\mathbf{x}}_d$.*

By Proposition .2, BWCP can adjust activation probability by changing the values of $\gamma_c$ and $\beta_c$ in Proposition 1 through BW module (see detail in Appendix A.4). Here we introduce a small positive constant $\delta$ to avoid the small activation feature value. To see how BW changes the activation probability of different channels, we consider two cases as shown in Proposition 3.

**Case 1:** $\beta_c \le 0$ and $\gamma_c \to 0$. In this case, the $c$-th channel of the BN layer would be activated with a small activation probability as it sufficiently approaches zero. We can see from Proposition 3, the activation probability of $c$-th channel still approaches zero after BW is applied, showing that the proposed BW module can keep the unimportant channels unchanged in this case. **Case 2:** $|\gamma_c| > 0$. For this case, the $c$-th channel of the BN layer would be activated with a high activation probability. From Proposition 3, the activation probability of $c$-th channel is enlarged after BW is applied. Therefore, our proposed BW module can increase the activation probability of important channels. Detailed proof of Proposition 3 can be found in Appendix A.5. We also empirically verify Proposition 3 in Sec. 5.3. Notice that we neglect a trivial case in which the channel can be also activated (*i.e.* $\beta_c > 0$ and $|\gamma_c| \to 0$). In fact, the channels can be removed in this case because the channel feature is always constant which can be deemed as a bias.

## 4.3 Soft Sampling Module

The soft sampling procedure samples the output of BW through a set of differentiable masks. To be specific, as shown in Fig.2, we leverage the Gumbel-Softmax sampling (Jang et al., 2017) that takes the activation probability generated by BW and produces a soft mask as given by

$$m_c = \mathrm{GumbelSoftmax}(P(\hat{X}_c > 0); \tau) \quad (6)$$

where $\tau$ is the temperature. By Eqn.(2) and Eqn.(6), BWCP stochastically prunes unimportant channels with activation probability. A smaller activation probability makes $m_c$ more likely to be

close to 0. Hence, our proposed BW can help identify less important channels by enlarging the activation probability of important channels, as mentioned in Sec.4.2. Note that $m_c$ can converge to 0-1 mask when $\tau$ approaches to 0. In the experiment, we find that setting $\tau = 0.5$ is enough for BWCP to achieve hard pruning at test time.

**Proposition 3** *Let* $\delta = \|\boldsymbol{\gamma}\|_2 \sqrt{\sum_{j=1}^C (\gamma_j \beta_c - \gamma_c \beta_j)^2 \rho_{cj}^2} / (\|\boldsymbol{\gamma}\|_2^2 - \sum_{j=1}^C \gamma_j^2 \rho_{cj})$. *With* $\hat{\gamma}$ *and* $\hat{\beta}$ *defined in Proposition 2, we have (1)* $P(\hat{X}_c > \delta) = 0$ *if* $|\gamma_c| \to 0$ *and* $\beta_c \leq 0$, *and (2)* $P(\hat{X}_c > \delta) \geq P(\tilde{X}_c \geq \delta)$ *if* $|\gamma_c| > 0$.

**Solution to residual issue.** Note that the number of channels in the last convolution layer must be the same as previous blocks due to the element-wise summation in the recent advanced CNN architectures (He et al., 2016; Huang et al., 2017). We solve this problem by letting BW layer in the last convolution layer and shortcut share the same mask as discussed in Appendix A.6.

## 4.4 TRAINING OF BWCP

This section introduces a sparsity regularization, which makes the model compact, and then describes the training algorithm of BWCP.

**Sparse Regularization.** With Proposition.1, we see a main characteristic of pruned channels in BN layer is that $\gamma_c$ sufficiently approaches 0, and $\beta_c$ is negative. By Proposition 3, we find that it is also a necessary condition that a channel can be pruned after BW module is applied. Hence, we obtain unnecessary channels by directly imposing a regularization on $\gamma_c$ and $\beta_c$ as given by

$$\mathcal{L}_{\text{sparse}} = \sum_{c=1}^C \lambda_1 |\gamma_c| + \lambda_2 \beta_c \qquad (7)$$

where the first term makes $\gamma_c$ small, and the second term encourages $\beta_c$ to be negative. The above sparse regularizer is imposed on all BN layers of the network. By changing the strength of regularization (*i.e.* $\lambda_1$ and $\lambda_2$), we can achieve different pruning ratios. In fact, $\beta_c$ and $|\gamma_c|$ represent the mean and standard deviation of a Normal distribution, respectively. Following the empirical rule of Normal distribution, setting $\lambda_1$ as triple or double $\lambda_2$ would be a good choice to encourage sparse channels in implementation. Moreover, we observe that 42.2% and 41.3% channels with $\beta_c \leq 0$, while 0.47% and 5.36% channels with $|\gamma_c| < 0.05$ on trained plain ResNet-34 and ResNet-50. Hence, changing the strength of regularization on $\gamma_c$ will affect FLOPs more than that of $\beta_c$. If one wants to pursue a more compact model, increasing $\lambda_1$ is more effective than $\lambda_2$.

**Training Algorithm.** BWCP can be easily plugged into a CNN by modifying the traditional BN operations. Hence, the training of BWCP can be simply implemented in existing software platforms such as PyTorch and TensorFlow. In other words, the forward propagation of BWCP can be represented by Eqn.(2-3) and Eqn.(6), all of which define differentiable transformations. Therefore, our proposed BWCP can train and prune deep models in an end-to-end manner. Appendix A.7 also provides the explicit gradient back-propagation of BWCP. On the other hand, we do not introduce extra parameters to learn the pruning mask $m_c$. Instead, $m_c$ in Eqn.(6) is totally determined by the parameters in BN layers including $\boldsymbol{\gamma}$, $\boldsymbol{\beta}$ and $\boldsymbol{\Sigma}$. Hence, we can perform joint training of pruning mask $m_c$ and model parameters. The BWCP framework is provided in Algorithm 1 of Appendix Sec A.6

**Final architecture.** The final architecture is fixed at the end of training. During training, we use the Gumbel-Softmax procedure by Eqn.(6) to produce a soft mask. At test time, we instead use a hard 0-1 mask achieved by a sign function (*i.e.* $\text{sign}(P(\hat{X}_c > 0) > 0.5)$) to obtain the network's output. To make the inference stage stable, we use a sigmoid-alike transformation to make the activation probability approach 0 or 1 in training. By this strategy, we find that both the training and inference stage are stable and obtain a fixed compact model. After training, we obtain the final compact model by directly pruning channels with a mask value of 0. Therefore, our proposed BWCP does not need an extra fine-tuning procedure.

## 5 EXPERIMENTS

In this section, we extensively experiment with the proposed BWCP on CIFAR-10/100 and ImageNet. We show the advantages of BWCP in both recognition performance and FLOPs reduction comparing with existing channel pruning methods. We also provide an ablation study to analyze the proposed framework. The details of datasets and training configurations are provided in Appendix B.

Table 1: Performance comparison between our proposed approach BWCP and other methods on CIFAR-10. "Baseline Acc." and "Acc." denote the accuracies of the original and pruned models, respectively. "Acc. Drop" means the accuracy of the base model minus that of pruned models (smaller is better). "Channels ↓", "Model Size ↓", and "FLOPs ↓" denote the relative reductions in individual metrics compared to the unpruned networks (larger is better). '*' indicates the method needs a extra fine-tuning to recover performance. The best-performing results are highlighted in bold.

| Model | Method | Baseline Acc. (%) | Acc. (%) | Acc. Drop | Channels ↓ (%) | Model Size ↓ (%) | FLOPs ↓ (%) |
|---|---|---|---|---|---|---|---|
| ResNet-56 | DCP* (Zhuang et al., 2018) | 93.80 | 93.49 | 0.31 | - | 49.24 | 50.25 |
| | AMC* (He et al., 2018b) | 92.80 | 91.90 | 0.90 | - | - | 50.00 |
| | SFP (He et al., 2018a) | 93.59 | 92.26 | 1.33 | 40 | – | **52.60** |
| | FPGM (He et al., 2019) | 93.59 | 92.93 | 0.66 | 40 | – | **52.60** |
| | SCP (Kang & Han, 2020) | 93.69 | 93.23 | 0.46 | **45** | 46.47 | 51.20 |
| | BWCP (Ours) | 93.64 | 93.37 | **0.27** | 40 | 44.42 | 50.35 |
| DenseNet-40 | Slimming* (Liu et al., 2017) | 94.39 | 92.59 | 1.80 | 80 | 73.53 | 68.95 |
| | Variational Pruning (Zhao et al., 2019) | 94.11 | 93.16 | 0.95 | 60 | 59.76 | 44.78 |
| | SCP (Kang & Han, 2020) | 94.39 | 93.77 | 0.62 | 81 | 75.41 | 70.77 |
| | BWCP (Ours) | 94.21 | 93.82 | **0.39** | 82 | **76.03** | **71.72** |
| VGGNet-16 | Slimming* (Liu et al., 2017) | 93.85 | 92.91 | 0.94 | 70 | 87.97 | 48.12 |
| | Variational Pruning (Zhao et al., 2019) | 93.25 | 93.18 | 0.07 | 62 | 73.34 | 39.10 |
| | SCP (Kang & Han, 2020) | 93.85 | 93.79 | 0.06 | 75 | 93.05 | 66.23 |
| | BWCP (Ours) | 93.85 | 93.82 | **0.03** | 76 | **93.12** | **68.08** |
| MobileNet-V2 | DCP* (Zhuang et al., 2018) | 94.47 | 94.69 | -0.22 | - | 23.6 | 27.0 |
| | MDP (Guo et al., 2020) | 95.02 | 95.14 | -0.12 | - | - | 28.7 |
| | BWCP (Ours) | 94.56 | 94.90 | **-0.36** | - | **32.3** | **37.7** |

## 5.1 RESULTS ON CIFAR-10

For CIFAR-10 dataset, we evaluate our BWCP on ResNet-56, DenseNet-40 and VGG-16 and compare our approach with Slimming (Liu et al., 2017), Variational Pruning (Zhao et al., 2019) and SCP (Kang & Han, 2020). These methods prune redundant channels using BN layers like our algorithm. We also compare BWCP with previous strong baselines such as AMC (He et al., 2018b) and DCP (Zhuang et al., 2018). The results of slimming are obtained from SCP (Kang & Han, 2020). As mentioned in Sec.4.2, our BWCP adjusts their activation probability of different channels. Therefore, it would present better recognition accuracy with comparable computation consumption by entirely exploiting important channels. As shown in Table 1, our BWCP achieves the lowest accuracy drops and comparable FLOPs reduction compared with existing channel pruning methods in all tested base networks. For example, although our model is not fine-tuned, the accuracy drop of the pruned network given by BWCP based on DenseNet-40 and VGG-16 outperforms Slimming with fine-tuning by 1.41% and 0.91% points, respectively. And ResNet-56 pruned by BWCP attains better classification accuracy than previous strong baseline DCP AMC (He et al., 2018b) and DCP (Zhuang et al., 2018) without an extra fine-tuning stage. Besides, our method achieves superior accuracy compared to the Variational Pruning even with significantly smaller model sizes on DensNet-40 and VGGNet-16, demonstrating its effectiveness. We also test BWCP with MobileNet-V2 on the CIFAR10 dataset. From Table 1, we see that BWCP achieves better classification accuracy while reducing more FLOPs We also report results of BWCP on CIFAR100 in Appendix B.3.

## 5.2 RESULTS ON IMAGENET

For ImageNet dataset, we test our proposed BWCP on two representative base models ResNet-34 and ResNet-50. The proposed BWCP is compared with SFP (He et al., 2018a)), FPGM (He et al., 2019), SSS (Huang & Wang, 2018)), SCP (Kang & Han, 2020) HRank (Lin et al., 2020) and DSA (Ning et al., 2020) since they prune channels without an extra fine-tuning stage. As shown in Table 2, we see that BWCP consistently outperforms its counterparts in recognition accuracy under comparable FLOPs. For ResNet-34, FPGM (He et al., 2019) and SFP (He et al., 2018a) without fine-tuning accelerates ResNet-34 by 41.1% speedup ratio with 2.13% and 2.09% accuracy drop respectively, but our BWCP without finetuning achieve almost the same speedup ratio with only 1.16% top-1 accuracy drop. On the other hand, BWCP also significantly outperforms FPGM (He et al., 2019) by 1.07% top-1 accuracy after going through a fine-tuning stage. For ResNet-50, BWCP still achieves better performance compared with other approaches. For instance, at the level of 40% FLOPs reduction, the top-1 accuracy of BWCP exceeds SSS (Huang & Wang, 2018) by 3.72%. Moreover, BWCP outperforms DSA (Ning et al., 2020) by top-1 accuracy of 0.34% and 0.21% at level of 40% and 50% FLOPs respectively. However, BWCP has slightly lower top-5 accuracy than DSA (Ning et al., 2020).

**Inference Acceleration.** We analyze the realistic hardware acceleration in terms of GPU and CPU running time during inference. The CPU type is Intel Xeon CPU E5-2682 v4, and the GPU is

Table 2: Performance of our proposed BWCP and other pruning methods on ImageNet using base models ResNet-34 and ResNet-50. '*' indicates the pruned model is fine-tuned.

| Model | Method | Baseline Top-1 Acc. (%) | Baseline Top-5 Acc. (%) | Top-1 Acc. Drop | Top-5 Acc. Drop | FLOPs ↓ (%) |
|---|---|---|---|---|---|---|
| | FPGM* (He et al., 2019) | 73.92 | 91.62 | 1.38 | 0.49 | 41.1 |
| | BWCP* (Ours) | 73.72 | 91.64 | **0.31** | **0.34** | **41.0** |
| ResNet-34 | SFP (He et al., 2018a) | 73.92 | 91.62 | 2.09 | 1.29 | 41.1 |
| | FPGM (He et al., 2019) | 73.92 | 91.62 | 2.13 | 0.92 | 41.1 |
| | BWCP (Ours) | 73.72 | 91.64 | **1.16** | **0.83** | **41.0** |
| | FPGM* (He et al., 2019) | 76.15 | 92.87 | 1.32 | 0.55 | **53.5** |
| | BWCP* (Ours) | 76.20 | 93.15 | **0.48** | **0.40** | 51.2 |
| | SSS (Huang & Wang, 2018) | 76.12 | 92.86 | 4.30 | 2.07 | 43.0 |
| | DSA (Ning et al., 2020) | – | – | 0.92 | 0.41 | 40.0 |
| | HRank* (Lin et al., 2020) | 76.15 | 92.87 | 1.17 | 0.64 | **43.7** |
| ResNet-50 | ThiNet* (Luo et al., 2017) | 72.88 | 91.14 | 0.84 | 0.47 | 36.8 |
| | BWCP (Ours) | 76.20 | 93.15 | **0.58** | **0.40** | 42.9 |
| | FPGM (He et al., 2019) | 76.15 | 92.87 | 2.02 | 0.93 | 53.5 |
| | SCP (Kang & Han, 2020) | 75.89 | 92.98 | 1.69 | 0.98 | **54.3** |
| | DSA (Ning et al., 2020) | – | – | 1.33 | 0.80 | 50.0 |
| | BWCP (Ours) | 76.20 | 93.15 | **1.02** | **0.60** | 51.2 |

Table 3: Effect of BW, Gumbel-Softmax (GS), and sparse Regularization in BWCP. The results are obtained by training ResNet-56 on CIFAR-10 dataset. 'BL' denotes baseline model.

| Cases | BW | GS | Reg | Acc. (%) | Model Size ↓ | FLOPs ↓ |
|---|---|---|---|---|---|---|
| BL | ✗ | ✗ | ✗ | 93.64 | - | - |
| (1) | ✓ | ✗ | ✗ | **94.12** | - | - |
| (2) | ✗ | ✗ | ✓ | 93.46 | - | - |
| (3) | ✗ | ✓ | ✓ | 92.84 | **46.37** | 51.16 |
| (4) | ✓ | ✓ | ✗ | 94.10 | 7.78 | 6.25 |
| (5) | ✓ | ✗ | ✓ | 92.70 | 45.22 | **51.80** |
| BWCP | ✓ | ✓ | ✓ | 93.37 | 44.42 | 50.35 |

Table 4: Effect of regularization strength $\lambda_1$ and $\lambda_2$ with magnitude $1e-4$ for the sparsity loss in Eqn.(7). The results are obtained using VGG-16 on CIFAR-100 dataset.

| $\lambda_1$ | $\lambda_2$ | Acc. (%) | Acc. Drop | FLOPs ↓ (%) |
|---|---|---|---|---|
| 1.2 | 0.6 | 73.85 | -0.34 | 33.53 |
| 1.2 | 1.2 | 73.66 | -0.15 | 35.92 |
| 1.2 | 2.4 | 73.33 | 0.18 | 54.19 |
| 0.6 | 1.2 | 74.27 | -0.76 | 30.67 |
| 2.4 | 1.2 | 71.73 | 1.78 | 60.75 |

NVIDIA GTX1080Ti. We evaluate the inference time using ResNet-50 with a mini-batch of 32 (1) on GPU (CPU). GPU inference batch size is larger than CPU to emphasize our method's acceleration on the highly parallel platform as a structured pruning method. We see that BWCP has 29.2% inference time reduction on GPU, from $48.7$ms for base ResNet-50 to $34.5$ms for pruned ResNet-50, and 21.2% inference time reduction on CPU, from $127.1$ms for base ResNet-50 to $100.2$ms for pruned ResNet-50.

## 5.3 ABLATION STUDY

**Effect of BWCP on activation probability.** From the analysis in Sec. 4.2, we have shown that BWCP can increase the activation probability of useful channels while keeping the activation probability of unimportant channels unchanged through BW technique. Here we demonstrate this using Resnet-34 and Resnet-50 trained on ImageNet dataset. We calculate the activation probability of channels of BN and BW layer. It can be seen from Fig.3 (a-d) that (1) BW increases the activation probability of important channels when $|\gamma_c| > 0$; (2) BW keeps the the activation probability of unimportant channels unchanged when $\beta_c \leq 0$ and $\gamma_c \to 0$. Therefore, BW indeed works by making useful channels more important and unnecessary channels less important, respectively. In this way, BWCP can identify unimportant channels reliably.

**Effect of BW, Gumbel-Softmax (GS) and sparse Regularization (Reg).** The proposed BWCP consists of three components including BW module (*i.e.* Eqn. (3)) and Soft Sampling module with Gumbel-Softmax (*i.e.* Eqn. (6)) and a spare regularization (*i.e.* Eqn. (7)). Here we investigate the effect of each component. To this end, five variants of BWCP are considered: (1) only BW module is used; (2) only sparse regularization is imposed; (3)BWCP w/o BW module; (4) BWCP w/o sparse regularization; and (5) BWCP with Gumbel-Softmax replaced by Straight Through Estimator (STE) (Bengio et al., 2013). For case (5), we select channels by hard 0-1 mask generated with $m_c = \text{sign}(P(\hat{X}_c > 0) - 0.5)$ [1]. The gradient is back-propagated through STE. From results on Table 3, we can make the following conclusions: **(a)** BW improves the recognition performance, implying that it can enhance the representation of channels; **(b)** sparse regularization on $\gamma$ and $\beta$ slightly harm the classification accuracy of original model but it encourages channels to be sparse as also shown in Proposition 3; **(c)** BWCP with Gumbel-Softmax achieves higher accuracy than STE, showing that a soft sampling technique is better than the deterministic ones as reported in (Jang et al., 2017).

---

[1] $y = \text{sign}(x) = 1$ if $x \geq 0$ and 0 if $x < 0$.

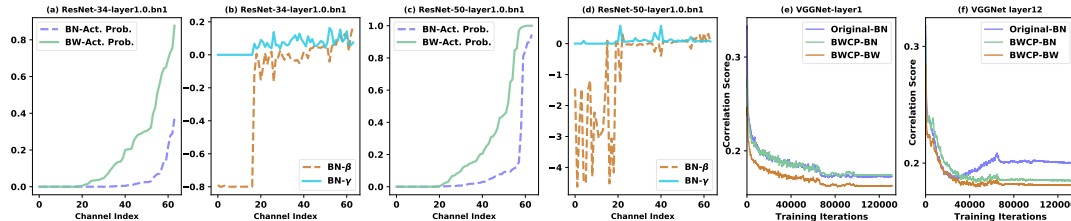

Figure 3: ((**a**) & (**b**)) and ((**c**) & (**d**)) show the effect of BWCP on activation probability with trained ResNet-34 and ResNet-50 on ImageNet, respectively. The proposed batch whitening (BW) can increase the activation probability of useful channels when $|\gamma_c| > 0$ while keeping the unimportant channels unchanged when when $\beta_c \leq 0$ and $\gamma_c \rightarrow 0$. (**e**) & (**f**) show the correlation score for the output response maps in shallow and deeper BWCP modules during the whole training period. BWCP has lower correlation score among feature channels than original BN baseline.

**Impact of regularization strength $\lambda_1$ and $\lambda_2$.** We analyze the effect of regularization strength $\lambda_1$ and $\lambda_2$ for sparsity loss on CIFAR-100. The trade-off between accuracy and FLOPs reduction is investigated using VGG-16. Table 4 illustrates that the network becomes more compact as $\lambda_1$ and $\lambda_2$ increase, implying that both terms in Eqn.(7) can make channel features sparse. Moreover, the flops metric is more sensitive to the regularization on $\gamma$, which validates our analysis in Sec.4.2). Besides, we should search for proper values for $\lambda_1$ and $\lambda_2$ to trade off between accuracy and FLOPs reduction, which is a drawback for our method.

**Effect of the number of BW.** Here the effect of the number of BW modules of BWCP is investigated trained on CIFAR-10 using Resnet-56 consisting of a series of bottleneck structures. Note that there are three BN layers in each bottleneck. We study four variants of BWCP: (a) we use BW to modify the last BN in each bottleneck module; hence there are a total of 18 BW layers in Resnet-56; (b) the last two BN layers are modified by our BW technique (36 BW layers) (c) All BN layers in bottlenecks are replaced by BW (54 BW layers), which is our proposed method. The results are reported in Table 5. We can see that BWCP achieves the best top-1 accuracy when BW acts on all BN layers, given the comparable FLOPs and model size. This indicates that the proposed BWCP more benefits from more BW layers in the network.

Table 5: Effect of the number of BW modules on CIFAR-10 dataset trained with ResNet-56. '# BW' indicates the number of BW. More BW modules in the network would lead to a lower recognition accuracy drop with comparable computation consumption.

| # BW | Acc. (%) | Acc. Drop | Model Size ↓ (%) | FLOPs ↓ (%) |
|------|----------|-----------|------------------|-------------|
| 18   | 93.01    | 0.63      | 44.70            | 50.77       |
| 36   | 93.14    | 0.50      | 45.29            | 50.45       |
| 54   | 93.37    | 0.27      | 44.42            | 50.35       |

**BWCP selects representative channel features.** It is worth noting that BWCP can whiten channel features after BN through BW as shwon in Eqn.(3). Therfore, BW can learn diverse channel features by reducing the correlations among channels(Yang et al., 2019b). We investigate this using VGGNet-16 with BN and the proposed BWCP trained on CIFAR-10. The correlation score can be calculated by taking the average over the absolute value of the correlation matrix of channel features. A larger value indicates that there is redundancy in the encoded features. We plot the correlation score among channels at different depths of the network. As shown in Fig.3 (e & f), channel features after BW block have significantly smaller correlations, implying that channels selected by BWCP are representative. This also accounts for the effectiveness of the proposed scheme.

## 6 DISCUSSION AND CONCLUSION

This paper presented an effective and efficient pruning technique, termed Batch Whitening Channel Pruning (BWCP). We show BWCP increases the activation probability of useful channels while keeping unimportant channels unchanged, making it appealing to pursue a compact model. Particularly, BWCP can be easily applied to prune various CNN architectures by modifying the batch normalization layer. However, to achieve different levels of FLOPs reduction, the proposed BWCP needs to search for the strength of sparse regularization. With probabilistic formulation in BWCP, the expected FLOPs can be modeled. The multiplier method can be used to encourage the model to attain target FLOPs. For future work, an advanced Pareto optimization algorithm can be designed to tackle such multi-objective joint minimization. We hope that the analyses of BWCP could bring a new perspective for future work in channel pruning.

**Ethics Statement.** We aim at compressing neural nets by the proposed BWCP framework. It could improve the energy efficiency of neural network models and reduce the emission of carbon dioxide. We notice that deep neural networks trained with BWCP can be plugged into portable or edge devices such as mobile phones. Hence, our work and AI in edge devices would have the same negative impact on ethics. Moreover, network pruning may have different effects on different classes, thus producing unfair models as a result. We will carefully investigate the results of our method on the fairness of the model output in the future.

**Reproducibility Statement.** For theoretical results, clear explanations of assumptions and a complete proof of propostion 1-3 are included in Appendix. To reproduce the experimental results, we provide training details and hyper-parameters in Appendix Sec.B. Moreover, we will also make our code available by a link to an anonymous repository during the discussion stage.

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

The appendix provides more details about approach and experiments of our proposed batch whitening channel pruning (BWCP) framework. The broader impact of this work is also discussed.

# A  MORE DETAILS ABOUT APPROACH

## A.1  CALCULATION OF COVARIANCE MATRIX $\boldsymbol{\Sigma}$

By Eqn.(1) in main text, the output of BN is $\tilde{x}_{ncij} = \gamma_c \bar{x}_{ncij} + \beta_c$. Hence, we have $\mathbb{E}[\tilde{\mathbf{x}}_c] = \frac{1}{NHW} \sum_{n,i,j}^{N,H,W} (\gamma_c \bar{x}_{ncij} + \beta_c) = \beta_c$. Then the entry in $c$-th row and $d$-th column of covariance matrix $\Sigma$ of $\tilde{\mathbf{x}}_c$ is calculated as follows:

$$\Sigma_{cd} = \frac{1}{NHW} \sum_{n,i,j}^{N,H,W} (\gamma_c \bar{x}_{ncij} + \beta_c - \mathbb{E}[\tilde{\mathbf{x}}_c])(\gamma_d \bar{x}_{ndij} + \beta_d - \mathbb{E}[\tilde{\mathbf{x}}_d]) = \gamma_c \gamma_d \rho_{cd} \qquad (8)$$

where $\rho_{cd}$ is the element in c-th row and j-th column of correlation matrix of $\bar{x}$. Hence, we have $\rho_{cd} \in [-1, 1]$. Furthermore, we can write $\Sigma$ into the vector form: $\boldsymbol{\Sigma} = \boldsymbol{\gamma}\boldsymbol{\gamma}^{\mathsf{T}} \odot \frac{1}{NHW} \sum_{n,i,j}^{N,H,W} \bar{\mathbf{x}}_{nij}\bar{\mathbf{x}}_{nij}^{\mathsf{T}} = \boldsymbol{\gamma}\boldsymbol{\gamma}^{\mathsf{T}} \odot \boldsymbol{\rho}$.

## A.2  PROOF OF PROPOSITION 1

For (1), we notice that we can define $\gamma_c = -\gamma_c$ and $\bar{X}_c = -\bar{X}_c \sim \mathcal{N}(0,1)$ if $\gamma_c < 0$. Hence, we can assume $\gamma_c > 0$ without loss of generality. Then, we have

$$
\begin{aligned}
P(Y_c > 0) = P(\tilde{X}_c > 0) &= P(\bar{X}_c > -\frac{\beta_c}{\gamma_c}) \\
&= \int_{-\frac{\beta_c}{\gamma_c}}^{+\infty} \frac{1}{\sqrt{2\pi}} \exp^{-\frac{t^2}{2}} dt \\
&= \int_{-\frac{\beta_c}{\gamma_c}}^{0} \frac{1}{\sqrt{2\pi}} \exp^{-\frac{t^2}{2}} dt + \int_{0}^{+\infty} \frac{1}{\sqrt{2\pi}} \exp^{-\frac{t^2}{2}} dt \\
&= \int_{0}^{\frac{\beta_c}{\gamma_c}} \frac{1}{\sqrt{2\pi}} \exp^{-\frac{t^2}{2}} dt + \int_{0}^{+\infty} \frac{1}{\sqrt{2\pi}} \exp^{-\frac{t^2}{2}} dt \\
&= \frac{\mathrm{Erf}(\frac{\beta_c}{\sqrt{2}\gamma_c}) + 1}{2}
\end{aligned}
\qquad (9)
$$

When $\gamma_c < 0$, we can set $\gamma_c = -\gamma_c$. Hence, we arrive at

$$P(Y_c > 0) = P(\tilde{X}_c > 0) = \frac{\mathrm{Erf}(\frac{\beta_c}{\sqrt{2}|\gamma_c|}) + 1}{2} \qquad (10)$$

For (2), let us denote $\bar{X}_c \sim \mathcal{N}(0,1)$, and $\tilde{X}_c = \gamma_c \bar{X}_c + \beta_c$ and $Y_c = \max\{0, \tilde{X}_c\}$ where $Y_c$ represents a random variables corresponding to output feature $\mathbf{y}_c$ in Eqn.(1) in main text. Firstly, it is easy to see that $P(\tilde{X}_c > 0) = 0 \Leftrightarrow \mathbb{E}_{\bar{X}_c}[Y_c] = 0$ and $\mathbb{E}_{\bar{X}_c}[Y_c^2] = 0$. In the following we show that $\mathbb{E}_{\bar{X}_c}[Y_c] = 0$ and $\mathbb{E}_{\bar{X}_c}[Y_c^2] = 0 \Leftrightarrow \beta_c \leq 0$ and $\gamma_c = 0$. Similar with (1), we assume $\gamma_c > 0$ without loss of generality.

For the sufficiency, we have

$$
\begin{aligned}
\mathbb{E}_{\bar{X}_c}[Y_c] &= \int_{-\infty}^{-\frac{\beta_c}{\gamma_c}} 0 \cdot \frac{1}{\sqrt{2\pi}} \exp^{-\frac{\bar{x}_c^2}{2}} d\bar{x}_c + \int_{-\frac{\beta_c}{\gamma_c}}^{+\infty} (\gamma_c \bar{x}_c + \beta_c) \cdot \frac{1}{\sqrt{2\pi}} \exp^{-\frac{\bar{x}_c^2}{2}} d\bar{x}_c, \\
&= \frac{\gamma_c \exp^{-\frac{\beta_c^2}{2\gamma_c^2}}}{\sqrt{2\pi}} + \frac{\beta_c}{2}(1 + \mathrm{Erf}[\frac{\beta_c}{\sqrt{2}\gamma_c}]),
\end{aligned}
\qquad (11)
$$

where $\mathrm{Erf}[x] = \frac{2}{\sqrt{\pi}} \int_0^x \exp^{-t^2} dt$ is the error function. From Eqn.(11), we have

$$\lim_{\gamma_c \to 0^+} \mathbb{E}_{\bar{X}_c}[Y_c] = \lim_{\gamma_c \to 0^+} \frac{\gamma_c \exp^{-\frac{\beta_c^2}{2\gamma_c^2}}}{\sqrt{2\pi}} + \lim_{\gamma_c \to 0^+} \frac{\beta_c}{2}(1 + \mathrm{Erf}[\frac{\beta_c}{\sqrt{2}\gamma_c}]) = 0 \qquad (12)$$

Table 6: Runing time comparison during training between BWCP, vanilla BN and SCP. The proposed BWCP achieves better trade-off between FLOPs reduction and accuracy drop although it introduces a little extra computational cost during training. 'F' denotes forward running time (s) while 'F+B' denotes forward and backward running time (s). The results are averaged over 100 iterations. The GPU is NVIDIA GTX1080Ti. The CPU type is Intel Xeon E5-2682 v4.

| Model | Mothod | CPU (F) (s) | CPU (F+B) (s) | GPU (F) (s) | GPU (F+B) (s) | Acc. Drop | FLOPs↓ (%) |
|---|---|---|---|---|---|---|---|
| | vanilla BN | 0.184 | 0.478 | 0.015 | 0.031 | 0 | 0 |
| ResNet-50 | SCP | 0.193 | 0.495 | 0.034 | 0.067 | 1.69 | 54.3 |
| | BWCP (Ours) | 0.239 | 0.610 | 0.053 | 0.104 | 1.02 | 51.2 |

In the same way, we can calculate

$$
\mathbb{E}_{\bar{x}_c}[Y_c^2] = \int_{-\infty}^{-\frac{\beta_c}{\gamma_c}} 0 \cdot \frac{1}{\sqrt{2\pi}} \exp^{-\frac{\bar{x}_c^2}{2}} d\bar{x}_c + \int_{-\frac{\beta_c}{\gamma_c}}^{+\infty} (\gamma_c \bar{x}_c + \beta_c)^2 \cdot \frac{1}{\sqrt{2\pi}} \exp^{-\frac{\bar{x}_c^2}{2}} d\bar{x}_c,
$$
$$
= \frac{\gamma_c \beta_c \exp^{-\frac{\beta_c^2}{2\gamma_c^2}}}{\sqrt{2\pi}} + \frac{\gamma_c^2 + \beta_c^2}{2} (1 + \mathrm{Erf}[\frac{\beta_c}{\sqrt{2}\gamma_c}]),
$$
(13)

From Eqn.(13), we have

$$
\lim_{\gamma_c \to 0^+} \mathbb{E}_{\bar{x}_c}[Y_c^2] = \lim_{\gamma_c \to 0^+} \frac{\gamma_c \beta_c \exp^{-\frac{\beta_c^2}{2\gamma_c^2}}}{\sqrt{2\pi}} + \lim_{\gamma_c \to 0^+} \frac{\gamma_c^2 + \beta_c^2}{2} (1 + \mathrm{Erf}[\frac{\beta_c}{\sqrt{2}\gamma_c}]) = 0
$$
(14)

For necessity, we show that if $\mathbb{E}_{\bar{x}_c}[Y_c] = 0$ and $\mathbb{E}_{\bar{x}_c}[Y_c^2] = 0$, then $\gamma_c \to 0$ and $\beta_c \le 0$. In essence, It can be acquired by solving Eqn Eqn.(11) and Eqn.(13). To be specific, $\beta_c * \mathrm{Eqn}.(.11) - \mathrm{Eqn}.(13)$ gives us $\gamma_c = 0^+$. Substituting it into Eqn.(.11), we can obtain $\beta_c \le 0$. This completes the proof.

## A.3 TRAINING OVERHEAD OF BWCP

The proposed BWCP introduces a little extra computational cost during training. To see this, we evaluate the computational complexity of SCP and BWCP for ResNet50 on ImageNet with an input image size of $224 \times 224$. We can see from the table below that the training BWCP is slightly slower on both CPU and GPU than the plain ResNet and SCP. In fact, the computational burden mainly comes from calculating the covariance matrix and its root inverse. In our paper, we calculate the root inverse of the covariance matrix by Newton's iteration, which is fast and efficient. Although BWCP brings extra training overhead, it achieves better top-1 accuracy drop under the same FLOPs consumption.

## A.4 PROOF OF PROPOSITION 2

First, we can derive that $\hat{X}_c = \boldsymbol{\Sigma}_N^{-\frac{1}{2}}(\boldsymbol{\gamma} \odot \bar{X} + \boldsymbol{\beta}) = \boldsymbol{\Sigma}_N^{-\frac{1}{2}}(\boldsymbol{\gamma} \odot \bar{X}) + \boldsymbol{\Sigma}_N^{-\frac{1}{2}}\boldsymbol{\beta} = (\boldsymbol{\Sigma}_N^{-\frac{1}{2}}\boldsymbol{\gamma}) \odot \bar{X} + \boldsymbol{\Sigma}_N^{-\frac{1}{2}}\boldsymbol{\beta}$.

Hence, the newly defined scale and bias parameters are $\hat{\boldsymbol{\gamma}} = \boldsymbol{\Sigma}_N^{-\frac{1}{2}}\boldsymbol{\gamma}$ and $\hat{\boldsymbol{\beta}} = \boldsymbol{\Sigma}_N^{-\frac{1}{2}}\boldsymbol{\beta}$. When $T = 1$, we have $\boldsymbol{\Sigma}_N^{-\frac{1}{2}} = \frac{1}{2}(3\mathbf{I} - \boldsymbol{\Sigma}_N)$ by Eqn.(5) in main text. Hence we obtain,

$$
\hat{\boldsymbol{\gamma}} = \frac{1}{2}(3\mathbf{I} - \boldsymbol{\Sigma}_N)\boldsymbol{\gamma} = \frac{1}{2}(3\mathbf{I} - \frac{\boldsymbol{\gamma}\boldsymbol{\gamma}^\mathsf{T}}{\|\boldsymbol{\gamma}\|_2^2} \odot \boldsymbol{\rho})\boldsymbol{\gamma}
$$
$$
= \frac{1}{2}(3\boldsymbol{\gamma} - \frac{1}{\|\boldsymbol{\gamma}\|_2^2} \left[ \sum_{j=1}^{C} \gamma_1 \gamma_j \rho_{1j} \gamma_j, \cdots, \sum_{j=1}^{C} \gamma_C \gamma_j \rho_{Cj} \gamma_j \right]^\mathsf{T})
$$
$$
= \frac{1}{2} \left[ (3 - \sum_{j=1}^{C} \frac{\gamma_j^2 \rho_{1j}}{\|\boldsymbol{\gamma}\|_2^2})\gamma_1, \cdots, (3 - \sum_{j=1}^{C} \frac{\gamma_j^2 \rho_{Cj}}{\|\boldsymbol{\gamma}\|_2^2})\gamma_C \right]^\mathsf{T}
$$
(15)

Similarly, $\hat{\boldsymbol{\beta}}$ can be given by

$$
\begin{aligned}
\hat{\boldsymbol{\beta}} &= \frac{1}{2}(3\mathbf{I} - \boldsymbol{\Sigma})\boldsymbol{\beta} = \frac{1}{2}(3\mathbf{I} - \frac{\boldsymbol{\gamma}\boldsymbol{\gamma}^{\mathsf{T}}}{\|\boldsymbol{\gamma}\|_2^2} \odot \boldsymbol{\rho})\boldsymbol{\beta} \\
&= \frac{1}{2}(3\boldsymbol{\beta} - \frac{1}{\|\boldsymbol{\gamma}\|_2^2} \left[ \sum_{j=1}^{C} \gamma_1 \gamma_j \rho_{1j} \beta_j, \cdots, \sum_{j=1}^{C} \gamma_C \gamma_j \rho_{Cj} \beta_j \right]^{\mathsf{T}}) \\
&= \frac{1}{2} \left[ 3\beta_1 - (\sum_{j=1}^{C} \frac{\gamma_j \beta_j \rho_{1j}}{\|\boldsymbol{\gamma}\|_2^2})\gamma_1, \cdots, 3\beta_C - (\sum_{j=1}^{C} \frac{\gamma_j \beta_j \rho_{Cj}}{\|\boldsymbol{\gamma}\|_2^2})\gamma_C \right]^{\mathsf{T}}
\end{aligned}
\tag{16}
$$

Taking each component of vector Eqn.(15-16) gives us the expression of $\hat{\gamma}_c$ and $\hat{\beta}_c$ in Proposition 2.

## A.5 PROOF OF PROPOSITION 3

For (1), through Eqn.(15), we acquire $|\hat{\gamma}_c| = \frac{1}{2}|3 - \sum_{j=1}^{C} \frac{\gamma_j^2 \rho_{cj}}{\|\boldsymbol{\gamma}\|_2^2}||\gamma_c|$. Therefore, $|\hat{\gamma}_c| \to 0$ if $|\gamma_c| \to 0$. On the other hand, by Eqn.(16), we have $\hat{\beta}_c \approx \frac{1}{2}(3 - \frac{\gamma_c^2}{\|\boldsymbol{\gamma}\|_2^2})\beta_c < \beta_c \le 0$. Here we assume that $\rho_{cd} = 1$ if $c = d$ and 0 otherwise. Note that the assumption is plausible by Fig.3 (e & f) in main text from which we see that the correlation among channel features will gradually decrease during training. We also empirically verify these two conclusions by Fig.4. From Fig.4 we can see that $|\hat{\gamma}_c| \ge |\gamma_c|$ where the equality holds iff $|\gamma_c| = 0$, and $\hat{\beta}_c$ is larger than $\beta_c$ if $\beta_c$ is positive, and vice versa. By Proposition 1, we arrive at

$$
P(\hat{X}_c > \delta) < P(\hat{X}_c > 0) = 0
\tag{17}
$$

where the first '$>$' holds since $\delta$ is a small positive constant and '$=$' follows from $|\hat{\gamma}_c| \to 0$ and $\hat{\beta}_c \le 0$. For (2), to show $P(\hat{X}_c > \delta) > P(\tilde{X}_c > \delta)$, we only need to prove $P(\bar{X}_c > \frac{\delta - \hat{\beta}_c}{|\hat{\gamma}_c|}) > P(\bar{X}_c > \frac{\delta - \beta_c}{|\gamma_c|})$, which is equivalent to $\frac{\delta - \hat{\beta}_c}{|\hat{\gamma}_c|} < \frac{\delta - \beta_c}{|\gamma_c|}$. To this end, we calculate

$$
\begin{aligned}
\frac{|\gamma_c|\hat{\beta}_c - |\hat{\gamma}_c|\beta_c}{|\hat{\gamma}_c| - |\gamma_c|} &= \frac{|\gamma_c|\frac{1}{2}(3\beta_c - (\sum_{j=1}^{C} \frac{\gamma_j \beta_j \rho_{cj}}{\|\boldsymbol{\gamma}\|_2^2})\gamma_c) - \frac{1}{2}(3 - \sum_{j=1}^{C} \frac{\gamma_j^2 \rho_{cj}}{\|\boldsymbol{\gamma}\|_2^2})|\gamma_c|\beta_c}{\frac{1}{2}(3 - \sum_{j=1}^{C} \frac{\gamma_j^2 \rho_{cj}}{\|\boldsymbol{\gamma}\|_2^2})|\gamma_c| - |\gamma_c|} \\
&= \frac{\sum_{j=1}^{C} \frac{\gamma_j \beta_j \gamma_c \rho_{cj}}{\|\boldsymbol{\gamma}\|_2^2} - \sum_{j=1}^{C} \frac{\gamma_j^2 \beta_c \rho_{cj}}{\|\boldsymbol{\gamma}\|_2^2}}{1 - \sum_{j=1}^{C} \frac{\gamma_j^2 \rho_{cj}}{\|\boldsymbol{\gamma}\|_2^2}} \\
&= \frac{\sum_{j=1}^{C} \frac{\gamma_j (\beta_j \gamma_c - \gamma_j \beta_c) \rho_{cj}}{\|\boldsymbol{\gamma}\|_2^2}}{1 - \sum_{j=1}^{C} \frac{\gamma_j^2 \rho_{cj}}{\|\boldsymbol{\gamma}\|_2^2}} \le \frac{\frac{1}{\|\boldsymbol{\gamma}\|_2}\sqrt{\sum_{j=1}^{C}(\beta_j \gamma_c - \gamma_j \beta_c)^2 \rho_{cj}^2}}{1 - \sum_{j=1}^{C} \frac{\gamma_j^2 \rho_{cj}}{\|\boldsymbol{\gamma}\|_2^2}} \\
&= \frac{\|\boldsymbol{\gamma}\|_2 \sqrt{\sum_{j=1}^{C}(\beta_j \gamma_c - \gamma_j \beta_c)^2 \rho_{cj}^2}}{\|\boldsymbol{\gamma}\|_2^2 - \sum_{j=1}^{C} \gamma_j^2 \rho_{cj}} = \delta
\end{aligned}
\tag{18}
$$

where the '$\le$' holds due to the Cauchy–Schwarz inequality. By Eqn.(18), we derive that $|\gamma_c|(\delta - \hat{\beta}_c) \le |\hat{\gamma}_c|(\delta - \beta_c)$ which is exactly what we want. Lastly, we empirically verify that $\delta$ defined in Proposition 3 is a small positive constant. In fact, $\delta$ represents the minimal activation feature value (i.e. $\hat{X}_c = \hat{\gamma}_c \bar{X}_c + \hat{\beta}_c \ge \delta$ by definition). We visualize the value of $\delta$ in shallow and deep layers in ResNet-34 during the whole training stage and value of $\delta$ of each layer in trained ResNet-34 on ImageNet dataset in Fig.5. As we can see, $\delta$ is always a small positive number in the training process. We thus empirically set $\delta$ as 0.05 in all experiments.

To conclude, by Eqn.(17), BWCP can keep the activation probability of unimportant channel unchanged; by Eqn.(18), BWCP can increase the activation probability of important channel. In this way, the proposed BWCP can pursue a compact deep model with good performance.

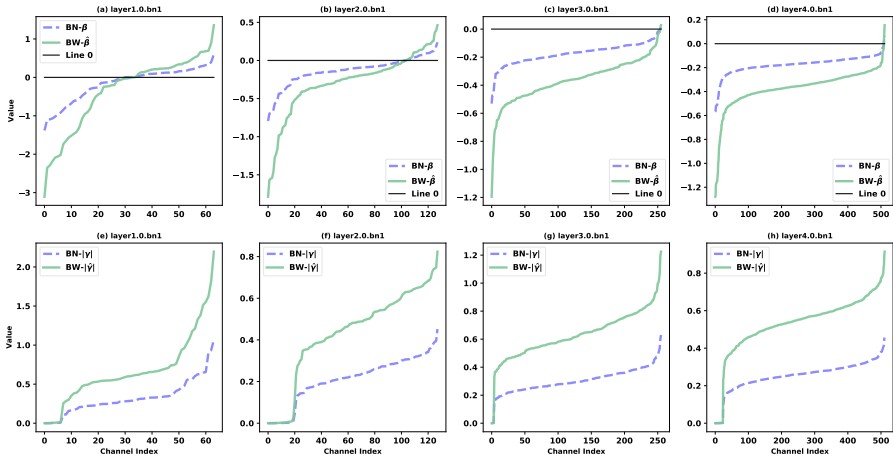

Figure 4: Experimental observation of how our proposed BWCP changes the values of $\gamma_c$ and $\beta_c$ in BN layers through the proposed BW technique. Results are obtained by tranining ResNet50 on ImageNet dataset. We investigate $\gamma_c$ and $\beta_c$ at different depths of the network including layer1.0.bn1, layer2.0.bn1,layer3.0.bn1 and layer4.0.bn1. **(a-d)** show BW enlarges $\beta_c$ when $\beta_c > 0$ while reducing $\beta_c$ when $\beta_c \le 0$. **(e-h)** show that BW consistently increases the magnitude of $\gamma_c$ across the network.

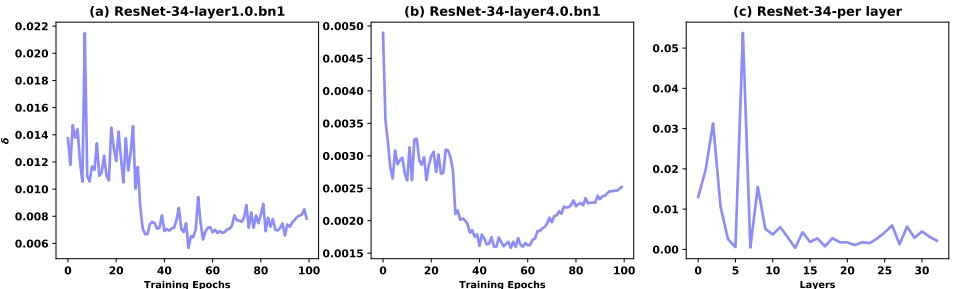

Figure 5: Experimental observation of the values of $\delta$ defined in proposition 3. Results are obtained by tranining ResNet-34 on ImageNet dataset. (**a** & **b**) investigate $\delta$ at different depths of the network including layer1.0.bn1 and layer4.0.bn1 respectively. (**c**) visualizes $\delta$ for each layer of ResNet-34. We see that $\delta$ in proposition 3 is always a small positive constant.

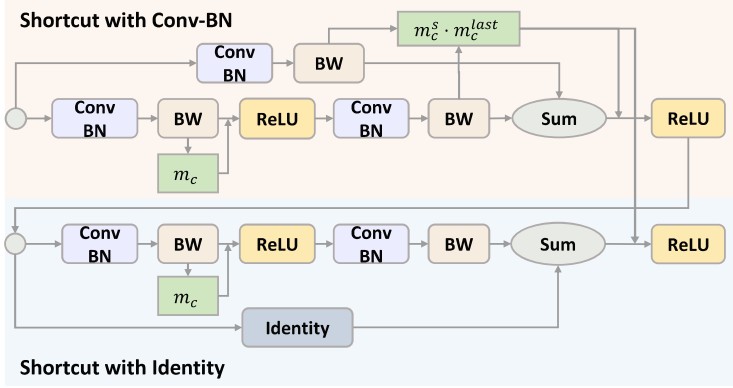

Figure 6: Illustration of BWCP with shortcut in basic block structure of ResNet. For shortcut with Conv-BN modules, we use a simple strategy that lets BW layer in the last convolution layer and shortcut share the same mask. For shortcut with identity mappings, we use the mask in previous layer.

---

**Algorithm 1** Forward Propagation of the proposed BWCP.

---

1: **Input**: mini-batch inputs $\mathbf{x} \in \mathbb{R}^{N \times C \times H \times W}$.
2: **Hyperparameters**: momentum $g$ for calculating root inverse of covariance matrix, iteration number $T$.
3: **Output**: the activations $\mathbf{x}^{out}$ obtained by BWCP.
4: calculate standardized activation: $\{\bar{\mathbf{x}}_c\}_{c=1}^C$ in Eqn.(1).
5: calculate the output of BN layer: $\tilde{\mathbf{x}}_c = \gamma_c \bar{\mathbf{x}}_c + \beta_c$.
6: calculate normalized covariance matrix: $\mathbf{\Sigma}_N = \frac{\gamma\gamma^\mathsf{T}}{\|\gamma\|_2^2} \odot \frac{1}{NHW} \sum_{n,i,j=1}^{N,H,W} \bar{\mathbf{x}}_{nij} \bar{\mathbf{x}}_{nij}^\mathsf{T}$
7: $\mathbf{\Sigma}_0 = \mathbf{I}$.
8: **for** $k = 1 \ to \ T$ **do**
9: $\quad \mathbf{\Sigma}_k = \frac{1}{2}(3\mathbf{\Sigma}_{k-1} - \mathbf{\Sigma}_{k-1}^3 \mathbf{\Sigma}_N)$
10: **end for**
11: calculate whitening matrix for training: $\mathbf{\Sigma}_N^{-\frac{1}{2}} = \mathbf{\Sigma}_T$.
12: calculate whitening matrix for inference: $\hat{\mathbf{\Sigma}}_N^{-\frac{1}{2}} \leftarrow (1-g)\hat{\mathbf{\Sigma}}_N^{-\frac{1}{2}} + g\mathbf{\Sigma}_N^{-\frac{1}{2}}$.
13: calculate whitened output: $\hat{\mathbf{x}}_{nij} = \mathbf{\Sigma}_N^{-\frac{1}{2}} \tilde{\mathbf{x}}_{nij}$.
14: calculate equivalent scale and bias defined by BW: $\hat{\gamma} = \mathbf{\Sigma}^{-\frac{1}{2}}\gamma$ and $\hat{\beta} = \mathbf{\Sigma}^{-\frac{1}{2}}\beta$.
15: calculate the activation probability by Proposition 2 with $\hat{\gamma}$ and $\hat{\beta}$, obtain soft masks $\{m_c\}_{c=1}^C$ by Eqn.(6).
16: calculate the output of BWCP: $\mathbf{x}_c^{out} = \hat{\mathbf{x}}_c \odot m_c$.

---

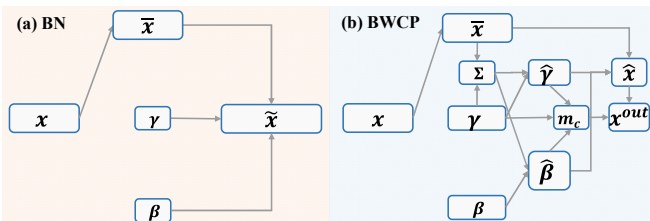

Figure 7: Illustration of forward propagation of **(a)** BN and **(b)** BWCP. The proposed BWCP prunes CNNs by replacing original BN layer with BWCP module.

## A.6 SOLUTION TO RESIDUAL ISSUE

The recent advanced CNN architectures usually have residual blocks with shortcut connections (He et al., 2016; Huang et al., 2017). As shown in Fig.6, the number of channels in the last convolution layer must be the same as in previous blocks due to the element-wise summation. Basically, there are two types of residual connections, *i.e.* shortcut with downsampling layer consisting of Conv-BN modules, and shortcut with identity. For shortcut with Conv-BN modules, the proposed BW technique is utilized in downsampling layer to generate pruning mask $m_c^s$. Furthermore, we use a simple strategy that lets BW layer in the last convolution layer and shortcut share the same mask as given by $m_c = m_c^s \cdot m_c^{last}$ where $m_c^{last}$ and $m_c^s$ denote masks of the last convolution layer. For shortcut with identity mappings, we use the mask in the previous layer. In doing so, their activated output channels must be the same.

## A.7 BACK-PROPAGATION OF BWCP

The forward propagation of BWCP can be represented by Eqn.(3-4) and Eqn.(9) in the main text (see detail in Table 1), all of which define differentiable transformations. Here we provide the back-propagation of BWCP. By comparing the forward representation of BN and BWCP in Fig.7, we need to back-propagate the gradient $\frac{\partial \mathcal{L}}{\partial \mathbf{x}_{nij}^{out}}$ to $\frac{\partial \mathcal{L}}{\partial \bar{\mathbf{x}}_{nij}}$ for backward propagation of BWCP. For simplicity, we neglect the subscript '$nij$'.

By chain rules, we have

$$\frac{\partial \mathcal{L}}{\partial \bar{\mathbf{x}}} = \hat{\gamma} \odot \mathbf{m} \odot \frac{\partial \mathcal{L}}{\partial \mathbf{x}^{out}} + \frac{\partial \mathcal{L}}{\partial \bar{\mathbf{x}}}(\mathbf{\Sigma}_N^{-\frac{1}{2}}) \tag{19}$$

Table 7: Performance of our BWCP on different base models compared with other approaches on CIFAR-100 dataset.

| Model | Method | Baseline Acc. (%) | Acc. (%) | Acc. Drop | Channels ↓ (%) | Model Size ↓ (%) | FLOPs ↓ (%) |
|---|---|---|---|---|---|---|---|
| | Slimming* (Liu et al., 2017) | 77.24 | 74.52 | 2.72 | **60** | **29.26** | **47.92** |
| ResNet-164 | SCP (Kang & Han, 2020) | 77.24 | 76.62 | 0.62 | 57 | 28.89 | 45.36 |
| | BWCP (Ours) | 77.24 | 76.77 | **0.47** | 41 | 21.58 | 39.84 |
| | Slimming* (Liu et al., 2017) | 74.24 | 73.53 | 0.71 | **60** | 54.99 | **50.32** |
| DenseNet-40 | Variational Pruning (Zhao et al., 2019) | 74.64 | 72.19 | 2.45 | 37 | 37.73 | 22.67 |
| | SCP (Kang & Han, 2020) | 74.24 | 73.84 | 0.40 | **60** | 55.22 | 46.25 |
| | BWCP (Ours) | 74.24 | 74.18 | **0.06** | 54 | 53.53 | 40.40 |
| VGGNet-19 | Slimming* (Liu et al., 2017) | 72.56 | 73.01 | -0.45 | **50** | 76.47 | 38.23 |
| | BWCP (Ours) | 72.56 | 73.20 | **-0.64** | 23 | 41.00 | 22.09 |
| | Slimming* (Liu et al., 2017) | 73.51 | 73.45 | 0.06 | **40** | 66.30 | 27.86 |
| VGGNet-16 | Variational Pruning (Zhao et al., 2019) | 73.26 | 73.33 | -0.07 | 32 | 37.87 | 18.05 |
| | BWCP (Ours) | 73.51 | 73.60 | **-0.09** | 34 | 58.16 | **34.46** |

where $\frac{\partial \mathcal{L}}{\partial \bar{\mathbf{x}}}(\boldsymbol{\Sigma}_N^{-\frac{1}{2}})$ denotes the gradient *w.r.t.* $\bar{\mathbf{x}}$ back-propagated through $\boldsymbol{\Sigma}_N^{-\frac{1}{2}}$. To calculate it, we first obtain the gradient *w.r.t.* $\boldsymbol{\Sigma}_N^{-\frac{1}{2}}$ as given by

$$\frac{\partial \mathcal{L}}{\partial \boldsymbol{\Sigma}_N^{-\frac{1}{2}}} = \gamma \frac{\partial \mathcal{L}}{\partial \hat{\gamma}}^{\mathsf{T}} + \beta \frac{\partial \mathcal{L}}{\partial \hat{\beta}}^{\mathsf{T}} \tag{20}$$

where

$$\frac{\partial \mathcal{L}}{\partial \hat{\gamma}} = \bar{\mathbf{x}} \odot \mathbf{m} \odot \frac{\partial \mathcal{L}}{\partial \mathbf{x}^{out}} + \frac{\partial \mathbf{m}}{\partial \hat{\gamma}} (\hat{\mathbf{x}} \odot \frac{\partial \mathcal{L}}{\partial \mathbf{x}^{out}}) \tag{21}$$

and

$$\frac{\partial \mathcal{L}}{\partial \hat{\beta}} = \mathbf{m} + \frac{\partial \mathbf{m}}{\partial \hat{\beta}} (\hat{\mathbf{x}} \odot \frac{\partial \mathcal{L}}{\partial \mathbf{x}^{out}}) \tag{22}$$

The remaining thing is to calculate $\frac{\partial \mathbf{m}}{\partial \hat{\gamma}}$ and $\frac{\partial \mathbf{m}}{\partial \hat{\beta}}$. Based on the Gumbel-Softmax transformation, we arrive at

$$\frac{\partial m_c}{\partial \hat{\gamma}_d} = \begin{cases} \frac{-m_c(1-m_c)f(\hat{\gamma}_c, \hat{\beta}_c)}{\tau P(\hat{X}_C > 0)(1 - P(\hat{X}_C > 0))} \frac{\beta_c \gamma_c}{|\gamma_c|^2} \text{ if } d = c \\ 0, \text{ otherwise} \end{cases} \tag{23}$$

$$\frac{\partial m_c}{\partial \hat{\beta}_d} = \begin{cases} \frac{m_c(1-m_c)f(\hat{\gamma}_c, \hat{\beta}_c)}{\tau P(\hat{X}_C > 0)(1 - P(\hat{X}_C > 0))}, \text{ if } d = c \\ 0, \text{ otherwise} \end{cases} \tag{24}$$

where $f(\hat{\gamma}_c, \hat{\beta}_c)$ is the probability density function of R.V. $\hat{X}_c$ as written in Eqn.(2) of main text.

To proceed, we deliver the gradient *w.r.t.* $\boldsymbol{\Sigma}_N^{-\frac{1}{2}}$ in Eqn.(20) to $\boldsymbol{\Sigma}$ by Newton Iteration in Eqn.(6) of main text. Note that $\boldsymbol{\Sigma}_N^{-\frac{1}{2}} = \Sigma_T$, we have

$$\frac{\partial \mathcal{L}}{\partial \boldsymbol{\Sigma}_N} = -\frac{1}{2} \sum_{k=1}^{T} (\boldsymbol{\Sigma}_{k-1}^3)^{\mathsf{T}} \frac{\partial \mathcal{L}}{\partial \boldsymbol{\Sigma}_k} \tag{25}$$

where $\frac{\partial \mathcal{L}}{\partial \boldsymbol{\Sigma}_k}$ can be calculated by following iterations:

$$\frac{\partial \mathcal{L}}{\partial \boldsymbol{\Sigma}_{k-1}} = \frac{3}{2} \frac{\partial \mathcal{L}}{\partial \boldsymbol{\Sigma}_k} - \frac{1}{2} \frac{\partial \mathcal{L}}{\partial \boldsymbol{\Sigma}_k} (\boldsymbol{\Sigma}_{k-1}^2 \boldsymbol{\Sigma})^{\mathsf{T}} - \frac{1}{2} (\boldsymbol{\Sigma}_{k-1}^2)^{\mathsf{T}} \frac{\partial \mathcal{L}}{\partial \boldsymbol{\Sigma}_k} \boldsymbol{\Sigma}^{\mathsf{T}}$$
$$- \frac{1}{2} (\boldsymbol{\Sigma}_{k-1})^{\mathsf{T}} \frac{\partial \mathcal{L}}{\partial \boldsymbol{\Sigma}_k} (\boldsymbol{\Sigma}_{k-1} \boldsymbol{\Sigma})^{\mathsf{T}} \, k = T, \cdots, 1.$$

Given the gradient *w.r.t.* $\boldsymbol{\Sigma}_N$ in Eqn.(25), we can calculate the gradient *w.r.t.* $\bar{\mathbf{x}}$ back-propagated through $\boldsymbol{\Sigma}_N^{-\frac{1}{2}}$ in Eqn.(19) as follows

$$\frac{\partial \mathcal{L}}{\partial \bar{\mathbf{x}}}(\boldsymbol{\Sigma}_N^{-\frac{1}{2}}) = (\frac{\gamma \gamma^{\mathsf{T}}}{\|\gamma\|^2} \odot (\frac{\partial \mathcal{L}}{\partial \boldsymbol{\Sigma}_N} + \frac{\partial \mathcal{L}}{\partial \boldsymbol{\Sigma}_N}^{\mathsf{T}})) \bar{\mathbf{x}} \tag{26}$$

Based on Eqn.(19-26), we obtain the back-propagation of BWCP.

## B  MORE DETAILS ABOUT EXPERIMENT

### B.1  DATASET AND METRICS

We evaluate the performance of our proposed BWCP on various image classification benchmarks, including CIFAR10/100 (Krizhevsky, 2009) and ImageNet (Russakovsky et al., 2015). The CIFAR-10 and CIFAR-100 datasets have 10 and 100 categories, respectively, while both contain 60k color images with a size of $32 \times 32$, in which 50k training images and 10K test images are included. Moreover, the ImageNet dataset consists of 1.28M training images and 50k validation images. Top-1 accuracy are used to evaluate the recognition performance of models on CIFAR 10/100. Top-1 and Top-5 accuracies are reported on ImageNet. We utilize the common protocols, *i.e.* number of parameters and Float Points Operations (FLOPs) to obtain model size and computational consumption.

For CIFAR-10/100, we use ResNet (He et al., 2016), DenseNet (Huang et al., 2017), and VGGNet (Simonyan & Zisserman, 2014) as our base model. For ImageNet, we use ResNet-34 and ResNet-50. We compare our algorithm with other channel pruning methods without a fine-tuning procedure. Note that a extra fine-tuning process would lead to remarkable improvement of performace (Ye et al., 2020). For fair comparison, we also fine-tune our BWCP to compare with those pruning methods. The training configurations are provided in Appendix B.2. The base networks and BWCP are trained together from scratch for all of our models.

### B.2  TRAINING CONFIGURATION

**Training Setting on ImageNet**. All networks are trained using 8 GPUs with a mini-batch of 32 per GPU. We train all the architectures from scratch for 120 epochs using stochastic gradient descent (SGD) with momentum 0.9 and weight decay 1e-4. We perform normal training without sparse regularization in Eqn.(7) on the original networks for first 20 epochs by following (Ning et al., 2020). The base learning rate is set to $0.1$ and is multiplied by $0.1$ after $50, 80$ and $110$ epochs. During fine-tuning, we use the standard SGD optimizer with Nesterov momentum $0.9$ and weight decay $0.00005$ to fine-tune pruned network for 150 epochs. We decay learning rate using cosine schedule with initial learning rate $0.01$. The coefficient of sparse regularization $\lambda_1$ and $\lambda_2$ are set to 7e-5 and 3.5e-5 to achieve Flops Reduction at a level of $40\%$, while $\lambda_1$ and $\lambda_2$ are set to 9e-5 and 3.5e-5 respectively to achieve FLOPs reduction at a level of $40\%$. Besides, the covariance matrix in the proposed BW technique is calculated within each GPU. Like (Huang et al., 2019), we also use group-wise decorrelation with group size 16 across the network to improve the efficiency of BW.

**Training setting on CIFAR-10 and CIFAR-100**. We train all models on CIFAR-10 and CIFAR-100 with a batch size of $64$ on a single GPU for 160 epochs with momentum 0.9 and weight decay 1e-4. The initial learning rate is 0.1 and is decreased by 10 times at $80$ and $120$ epoch. The coefficient of sparse regularization $\lambda_1$ and $\lambda_2$ are set to 4e-5 and 8e-5 for CIFAR-10 dataset and 7e-6 and 1.4e-5 for CIFAR-100 dataset.

### B.3  MORE RESULTS OF BWCP

The results of BWCP on CIFAR-100 dataset is reported in Table 7. As we can see, our approach BWCP achieves the lowest accuracy drops and comparable FLOPs reduction compared with existing channel pruning methods in all tested base models.

