# OpenReview forum: "BWCP: Probabilistic Learning-to-Prune Channels for ConvNets via Batch Whitening"
_ICLR.cc/2022/Conference — ICLR 2022 Submitted_

### Official Review · Reviewer_8cV6 · 2021-11-02

**Correctness:** 3
**Technical Novelty And Significance:** 3
**Empirical Novelty And Significance:** 3
**Recommendation:** 6
**Confidence:** 5

**Main Review:**

Feature whitening reduces the correlation between features. It was shown that the whitened representation facilitates the training. Another property is that the whitening also leads to a sparse representation. This paper shows a nice application of this property in training compact neural networks.

This direction is very important, neural networks have been dense and large. Whitening has been found to be the more optimal normalization  in terms of training of neural networks. However, its benefit in compressing is barely known to the community. This paper is a good attempt in this direction.

The downsides:

(1) Some reorganization could clarify the presentation. The reader may want to have a quick understanding of why whitening is chosen over standardization. Some plots similar to Fig. 10,11 in Ye et. al network deconvolution may be helpful. It will be beneficial to make the paper more self-contained, e.g. clearly formulate Gumble softmax  in eq 6.


(2) It is not clear whether the improvement is due to whitening or pruning. Whitening based methods such as decorrelated Batch Norm and IterNorm by Huang et. al generally lead to better performance compared to standard normalization methods. The benefits of sparsity may have already been exploited by these methods. If this is the case, one might be able to directly train a narrower network without the Gumbel softmax trick. Can the authors provide more insights/experiments to this? How much better is this method compared to direct whitened training?

a typo in page 8: spare -->sparse

**Summary Of The Paper:**

This paper proposes a way to train compact neural networks. The feature channels are whitened, then combined with a sampling method. At the testing time, this sampling is discarded and replaced with a deterministic selection.

The benefits of whitening in improving the convergence of NNs is widely explored. This paper explores another useful property of whitening,  the sparsity in the whitened results. Since the representation is sparser,  one might use fewer weights to reach similar performance. This paper presents such a solution.

**Summary Of The Review:**

The paper has room for improvements, but it is a nice new attempt to the pruning of networks.

---

### Official Review · Reviewer_PUWD · 2021-11-02

**Correctness:** 3
**Technical Novelty And Significance:** 2
**Empirical Novelty And Significance:** 1
**Recommendation:** 3
**Confidence:** 5

**Main Review:**

Strengths:

1.	They propose a probabilistic pruning method, quite different from existing pruning methods which act in a deterministic manner.
2.	Their method is shown effective on CIFAR-10/100 and ImageNet.

Weaknesses:

1.	Severe idea overlap.
The main spirit of this paper, “unimportant channels are stochastically pruned with activation probability”, namely the so-called “probabilistic” pruning, has been explored in the literature. See [*1], they proposed to assign activation probabilities to different filters; more important filters have larger activation probability.  This is very similar to what is proposed in this paper: “we assign each channel with an activation probability (i.e. the probability of a channel being activated)”, “A larger activation probability indicates that the corresponding channel is more likely to be preserved”. Please clarify how your method is substantially different from [*1] (if any).
Besides, there is no determined relationship between regularization strength and the final speedup in this method. Thus, “we should search for proper values for λ1 and λ2 to trade off between accuracy and FLOPs reduction, which is a drawback for our method”. However, in [*1], how many filters are imposed with probabilistic pruning is explicitly linked with the desired layer-wise sparsity, thus their method can achieve desired speedup for sure. In this sense, [*1] is also more convenient than the BWCP in practice.

2.	Performance is not strong. Missing important comparison with more recent methods.
In Tab. 2, ResNet50, ImageNet. They do not have any 2021 papers compared. If compared to the more recent 2021 methods (e.g., [*3-*4]), BWCP is actually not as competitive. For example, [*3] achieves 75.36% top-1 accuracy under speedup 2.31x, while BWCP only achieves 75.18% under even less speedup (2.05x).

3.	In terms of the proposed methodology itself, it is a BN-based pruning method. While not all deep CNNs have BN. For example, in image super-resolution (SR), BN is not used in SOTA SR network (such as [*2]). This limits the potential application of this method, while related works (such as [*1], HRank, DSA, [*3-*4], etc) do not have this limitation.

[*1] Structured Probabilistic Pruning for Convolutional Neural Network Acceleration, BMVC, 2018

[*2] Image super-resolution using very deep residual channel attention networks, ECCV, 2018

[*3] Neural pruning via growing regularization, ICLR, 2021.

[*4] Towards compact cnns via collaborative compression, CVPR, 2021


**Summary Of The Paper:**

This paper presents a probabilistic channel pruning method (BWCP) for accelerating CNNs. The key newly proposed technique is “batch whitening”. They evaluate their method on CIFAR-10/100 and ImageNet compared to other recent filter pruning methods.

**Summary Of The Review:**

There is a severe idea overlap between this work and previous work. Meanwhile, the empirical performance is not strong.

---

### Official Review · Reviewer_qyNE · 2021-11-02

**Correctness:** 4
**Technical Novelty And Significance:** 3
**Empirical Novelty And Significance:** 3
**Recommendation:** 5
**Confidence:** 4

**Main Review:**

Strengths:
1. Based on the proposed stochastic pruning scheme, the full channel space is preserved during the training process, which avoid stuck in some local minimum due to improper selection of channels to be pruned.
2. The batch whitening module, together with the regularization term on per-channel scaling factors and biases, automatically increases the activation probability of informative channels, so that unimportant channels can be identified more easily.
3. The effectiveness of the proposed method is validated by extensive experiments on various network architectures and ablation studies.

Weaknesses:
1. The training efficiency of the proposed method is not discussed. Will the stochastic pruning scheme require additional training epochs, compared with other channel pruning methods?
2. Section 4.4, Equation (7). As shown in Proposition 1, the activation probability is determined by $\beta_{c} / \left| \gamma_{c} \right|$, so it seems more reasonable to define the second regularization term as $\lambda_{2} \beta_{c} / \left| \gamma_{c} \right|$, rather than $\lambda_{2} \beta_{c}$ (or, is it due to the optimization difficulty of the former one?).
3. Section 4.4, last paragraph. Authors state that a sigmoid-like transformation is used to make activation probabilities approach 0 or 1 during training. How is this carried out exactly? It would be helpful to provide more technical details to improve the reproducibility of the proposed method.

**Summary Of The Paper:**

In this paper, authors propose to compress convolutional layers via batch whitening channel pruning (BWCP). By apply batch whitening on incoming feature maps, unimportant channels are automatically identified and stochastically discarded during the training process. A sigmoid-like transformation is used to push activation probabilities to either 0 or 1, so that additional fine-tuning is not needed, unlike many other channel pruning methods.

**Summary Of The Review:**

The training efficiency remains unclear. Although the stochastic pruning scheme can explore the channel space more sufficiently, it is also possible that more training time are needed for such exploration. More details should be provided here.

---

### Official Review · Reviewer_kp3m · 2021-11-02

**Correctness:** 2
**Technical Novelty And Significance:** 3
**Empirical Novelty And Significance:** 2
**Recommendation:** 5
**Confidence:** 4

**Main Review:**

The strength of this paper can be summarized into the following points.
1. This method has some good properties for pruning, like structural pruning, simultaneously pruning and training, and so on.
2. The analysis of whitening shows that the activation probability of important channels will become larger, which might be helpful for channel pruning.
3. The method generally achieves better results compared to similar methods.

The weakness of this paper are:
1. Batch-whitening is built on batch normalization, which may limit the application scope of this paper. In language modelings and vision transformers, layer normalization is widely used instead of batch normalization. Does whitening can also be applied to such settings? In addition, for downstream tasks, like object detection, the image resolution becomes larger, and the allowed mini-batch size will decrease dramatically. When the mini-batch size is limited, dose batch-whitening suffers from the poor estimation of $\Sigma$? In another word, the proposed method may perform poorly when batch normalization is unreliable.
2. Some arguments are not well supported by the experimental results. In the paper, the authors argue that 'bach-whitening increase the activation probability of useful channels while keeping the activation probability of unimportant channels unchanged'. This is demonstrated in Fig.3 (a.b). However, increasing the activation probability may not improve pruning. In Table. 3, Fig. the relative performance changes between case (3) and BL is similar to case (1) and BWCP. If increasing the activation probability is helpful, we should see BWCP achieves a more obvious performance gain compared to the case (3). The current ablation study does not support the argument that increasing the activation probability benefits pruning.
3. It seems that the batch-whitening process incurs additional computational costs since you need to keep updating the covariance matrix. Although newton iteration improves the efficiency, the training time is also increased. If we measure the training budget by the training time, other methods actually use a smaller training budget.
4. The experimental results are not very significant compared to previous methods. The improvement over previous baselines is around 0.3 for ResNet-50.

**Summary Of The Paper:**

The paper presents a probabilistic channel pruning method based on batch whitening to accelerate and compress CNNs.

**Summary Of The Review:**

In summary, I think this is a borderline paper. It has some good properties, but it is limited by using batch normalization. It also introduces additional costs at the training time.

---

### Decision · Program_Chairs · 2022-01-20

**Decision:**

Reject

**Comment:**

The reviewers consider the authors' approach to pruning of convolutional networks reasonable; but neither sufficiently novel nor sufficiently well explored for inclusion in the conference.  In particular, the reviewers would like to see a more explicit discussion of the effect on training time of the authors' method, and more discussion and comparison against previous probabilistic pruning methods.